# Highly efficient electrosynthesis of hydrogen peroxide on a superhydrophobic three-phase interface by natural air diffusion

Qizhan Zhang[1,2,3,4], Minghua Zhou [1,2,3,4✉], Gengbo Ren[1,2,3,4], Yawei Li[1,2,3,4], Yanchun Li[1,2,3,4] & Xuedong Du[1,2,3,4]

Hydrogen peroxide ($H_2O_2$) synthesis by electrochemical oxygen reduction reaction has attracted great attention as a green substitute for anthraquinone process. However, low oxygen utilization efficiency (<1%) and high energy consumption remain obstacles. Herein we propose a superhydrophobic natural air diffusion electrode (NADE) to greatly improve the oxygen diffusion coefficient at the cathode about 5.7 times as compared to the normal gas diffusion electrode (GDE) system. NADE allows the oxygen to be naturally diffused to the reaction interface, eliminating the need to pump oxygen/air to overcome the resistance of the gas diffusion layer, resulting in fast $H_2O_2$ production (101.67 mg h$^{-1}$ cm$^{-2}$) with a high oxygen utilization efficiency (44.5%–64.9%). Long-term operation stability of NADE and its high current efficiency under high current density indicate great potential to replace normal GDE for $H_2O_2$ electrosynthesis and environmental remediation on an industrial scale.

[1] Key Laboratory of Pollution Process and Environmental Criteria, Ministry of Education, College of Environmental Science and Engineering, Nankai University, 300350 Tianjin, China. [2] Tianjin Key Laboratory of Environmental Technology for Complex Trans-Media Pollution, Nankai University, 300350 Tianjin, China. [3] Tianjin Key Laboratory of Urban Ecology Environmental Remediation and Pollution Control, College of Environmental Science and Engineering, Nankai University, 300350 Tianjin, China. [4] Tianjin Advanced Water Treatment Technology International Joint Research Center, College of Environmental Science and Engineering, Nankai University, 300350 Tianjin, China. ✉email: zhoumh@nankai.edu.cn

H$_2$O$_2$ is a well-known clean oxidant with only water as its by-product, and is widely employed in chemical industries and environmental remediation[1,2]. Most H$_2$O$_2$ is industrially synthesized by the anthraquinone oxidation process, which requires a complicated operation, including hydrogenation, oxidation of anthrahydroquinone by O$_2$, and extraction and purification of H$_2$O$_2$[3]. High-energy consumption, substantial organic by-product waste, and transportation constraints are the main problems in the application of H$_2$O$_2$[4,5]. As an alternative, many efforts have been made to develop efficient and in situ synthesis methods. The oxygen reduction reaction (ORR), as an important green, cathodic process, can proceed by a direct two-electron reduction to produce H$_2$O$_2$, which has garnered great attention ascribed to its advantages, such as environmental friendliness and cost-effectiveness[6,7].

For those H$_2$O$_2$-based electrochemical advanced oxidation processes (EAOPs), e.g., electro-Fenton (EF) and photoelectro-Fenton (PEF), efficient H$_2$O$_2$ production is particularly important, which can promote the formation of hydroxyl radicals to degrade organic pollutants[8,9]. Catalyst selectivity, oxygen mass transfer, and electron transfer at the cathodic reaction interface are three important factors for achieving efficient H$_2$O$_2$ production[10]. Carbonaceous materials, including carbon black (CB), nanotubes, and graphene, have been proved to be good ORR catalysts for H$_2$O$_2$ synthesis[11,12]. As the applied potential increases, the electron transfer is enhanced. However, when it continues to increase to a certain extent, ORR will be controlled by oxygen mass transfer[13,14] due to the low solubility of oxygen in water at room temperature and pressure (~8 mg L$^{-1}$), leading to retarded kinetics of the two-electron ORR[15].

In recent years, some researchers are committed to solving these problems in various ways[16,17]. Gas diffusion electrode (GDE) is currently the most widely used strategy to improve the oxygen mass transfer efficiency of ORR, which allows oxygen to be supplied externally to the cathode surface without the need for its dissolution in the electrolyte and promotes H$_2$O$_2$ electrosynthesis efficiency[18]. Brillas et al. used GDE for electrochemical generation of H$_2$O$_2$ with pure O$_2$ at a flow rate of 60 mL min$^{-1}$, obtaining the optimal H$_2$O$_2$ production of 59 mg h$^{-1}$ cm$^{-2}$ at a current density of 242 mA cm$^{-2}$; however, only a small fraction of O$_2$ was utilized, and the GDE was rapidly damaged without injecting O$_2$[19].

Over the years, many researchers have focused on improving the catalytic layer of GDE to increase H$_2$O$_2$ production, but little attention has been paid to improve O$_2$ mass transfer efficiency[14,20–22]. Moreira et al. prepared GDE modified with 0.5% Sudan Red 7B, obtaining 8.9 mg h$^{-1}$ cm$^{-2}$ of H$_2$O$_2$ with a current efficiency of 17.87% when 0.3-bar O$_2$ was supplied at a current density of 75 mA cm$^{-2}$ [23]. It is noteworthy that the energy consumption was 118.0 kWh kg$^{-1}$ excluding the energy consumption of aeration. It has been well known that a higher O$_2$ supply promotes the production of H$_2$O$_2$[24], but the oxygen utilization efficiency (OUE) of GDEs is extremely low (usually <1%)[25,26]. Yu et al. proposed an improved GDE to synthesize H$_2$O$_2$, which increased the OUE to 8.84%, and the yield of H$_2$O$_2$ reached 11.16 mg h$^{-1}$ cm$^{-2}$ at the current density of 35.7 mA cm$^{-2}$. However, the aeration energy consumption (64.1 kWh kg$^{-1}$) was more than three times the electric energy consumption of H$_2$O$_2$ electrosynthesis (17.9 kWh kg$^{-1}$)[27]. When scaling up the H$_2$O$_2$-based EAOPs with GDEs, a significant energy loss occurs unavoidably[13]. Salmerón et al. devised a pilot unit based on a normal GDE (100 L) for electrosynthesis of H$_2$O$_2$, which caused the aeration energy consumption up to 1077 kWh kg$^{-1}$ (20 times the electric energy consumption of H$_2$O$_2$ electrosynthesis)[28]. Therefore, it is necessary to enhance oxygen mass transfer efficiency and OUE of normal GDEs, reduce the aeration energy consumption, and even produce H$_2$O$_2$ efficiently without aeration.

In addition, some successful modifications of cathodic catalytic interfaces have been explored to increase the two-electron ORR activity[29–31]. Yu et al. first used CB and polytetrafluoroethylene (PTFE) to develop a catalytic interface of ORR, and the yield of H$_2$O$_2$ was increased by about 10.7 times under the optimum PTFE to CB mass ratio of 5:1[32]. Xu et al. proposed using hydrophobic dimethyl silicon (DMS) oil layer to modify the cathode to enhance the surface waterproofing performance and improve oxygen transfer[33]. Though these methods seemed to be effective in improving performance, how to regulate the hydrophobicity of the catalytic layer to establish a stable superhydrophobic three-phase interface and how it affects the catalytic characteristics are still unclear.

In this work, we design a natural air diffusion electrode (NADE), which allows air to naturally diffuse to the ORR interface without additional aeration, demonstrating the highest performance in H$_2$O$_2$ production using carbon cathode to date and very high efficiency for organic pollutant degradation by EF and PEF. A carbon felt (CF) is used as matrix and diffusion layer, which not only simplifies the manufacturing process of normal GDE, but also enables atmospheric air to naturally diffuse into the catalytic layer. By regulating the hydrophobicity of the catalytic layer to establish a stable superhydrophobic three-phase interface, the relationship between the performance of H$_2$O$_2$ production and catalytic layer characteristics such as oxygen mass transport, electron transfer, and stability is established. Moreover, the oxygen mass transfer restriction in ORR, oxygen diffusion coefficient, and OUE of the cathode are carefully evaluated and compared with normal GDE. NADE eliminates aeration equipment and improves the H$_2$O$_2$ electrosynthesis efficiency, which is conducive to the application of EAOPs in the purification of organic wastewater.

## Results

**Fabrication and characterization of NADE.** As presented in Fig. 1, CF modified by PTFE acts as the gas diffusion layer and substrate, while CB is loaded on the other side to form NADE (detailed NADE fabrication is described in "Methods"). The CF consisted of 10-μm-diameter fibers forming an interconnected network with an interfiber distance of ~50 μm (Supplementary Fig. 1a) and the porosity was more than 90%; thereby, the oxygen mass transfer hindrance in the diffusion layer became extremely low, which rendered oxygen in the atmosphere actively diffuse to the reaction interface without air pumps[34]. After modification, CF had better strength and superhydrophobicity (water contact angle (CA) of 139°) while keeping high macroporosity favorable for oxygen diffusion (Fig. 1b and Supplementary Fig. 1b). The O$_2$ diffusion coefficient of the modified CF was established to be $1.15 \times 10^{-1}$ cm$^2$ s$^{-1}$ by the oxygen diffusion coefficient prediction model (Eqs. (9)–(12)), which was close to the reported value of oxygen diffusion coefficient in air ($2 \times 10^{-1}$ cm$^2$ s$^{-1}$)[35]. In contrast, microporous diffusion layer of traditional GDE is densely structured, and the pore structures for O$_2$ diffusion are very scarce (Supplementary Fig. 2), resulting in a low O$_2$ diffusion coefficient ($2.02 \times 10^{-2}$ cm$^2$ s$^{-1}$ calculated by the model, which was comparable with the measured values in the literature ($1 \times 10^{-2}$–$3.6 \times 10^{-2}$ cm$^2$ s$^{-1}$); Supplementary Fig. 3)[36–38]. Therefore, unlike traditional GDEs that require additional gas compression equipment, this NADE allowed the rapid and natural transport of O$_2$ from atmosphere to the cathode, which completely eliminated aeration energy consumption and was beneficial for cost-effective application[18,39].

In addition to the oxygen mass transfer efficiency of the diffusion layer, the hydrophobicity/hydrophilicity of the catalytic layer interface exerted significant effects on the cathode performance. A more hydrophobic cathode surface that favored the O$_2$ transfer to the reaction interface was responsible for fast ORR rates[40].

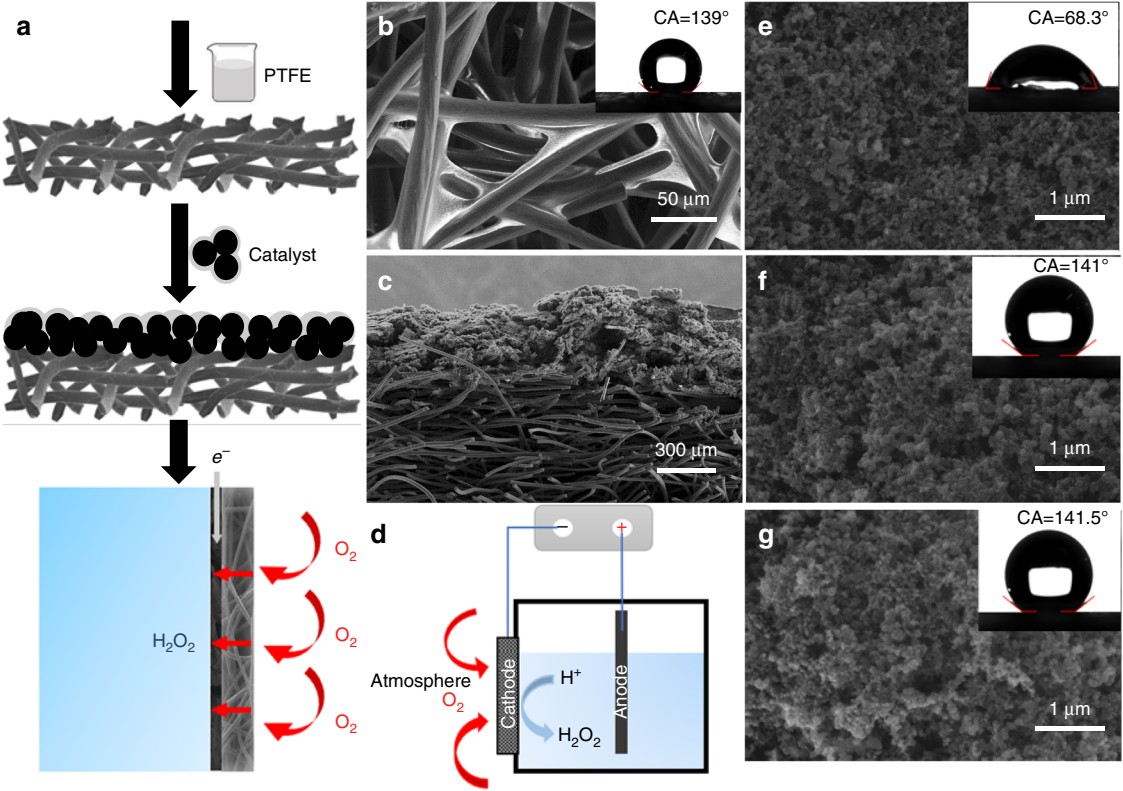

**Fig. 1 Schematic illustrations and SEM images of NADE. a** NADE fabrication process and schematic illustration of the three-phase electrocatalytic system. **b** SEM images and contact angle of modified carbon felt (inset). **c** SEM images of the modified carbon felt substrate coated with carbon black. **d** Schematic diagram of NADE electrocatalytic reactor; SEM images of catalytic layers with **e** PTFE/CB = 0.1, **f** PTFE/CB = 0.6, and **g** PTFE/CB = 1.5.

Through the interface CA measurement, different mass ratios of PTFE/CB were studied to construct a stable superhydrophobic interface. It showed that when the mass ratio of PTFE/CB was 0.1, the surface of the catalytic layer was hydrophilic (Fig. 1e; CA = 68.33°). Interestingly, when the mass ratio increased to 0.2, the interface began to exhibit a slight hydrophobicity (Supplementary Fig. 4; CA = 100.11°), but became hydrophilic (CA = 63.38°) after a few minutes under the influence of water droplets (Supplementary Fig. 5a). It meant that the interface was not waterproof and could not maintain the three-phase interface for a long time. As mentioned in many literatures, the three-phase interfaces on the cathode had a short life and were easily flooded in the electrolyte, resulting in the narrowing of oxygen transport channel and the decreasing of oxygen transport rate[30,33,41,42]. As the proportion of PTFE increased, the hydrophobicity of the interface was significantly enhanced. After the ratio was increased to 0.6, a stable superhydrophobic interface was formed (Fig. 1f; CA = 141.02°) and the interface basically did not change with time (Supplementary Fig. 5c). Even after the electrolysis for 20 h at the current density of 60 mA cm$^{-2}$, the hydrophobic interface of NADE could be maintained (Supplementary Fig. 6). The hydrophobicity of the catalytic layer did not change significantly as the amount of PTFE continued to increase. We systematically studied the influence of PTFE on hydrophobicity of the catalytic layer on CF and the change in the hydrophobic interface with time. Compared with the complex steps of adding a hydrophobic film on the surface of the catalytic layer in literature[30,33], we also obtained a stable superhydrophobic interface by simply regulating the amount of PTFE in the catalytic layer, and realized a long-term operation in the following experiments.

From the SEM images (Fig. 1e–g; Supplementary Fig. 7), it is clearly shown that the catalytic layer of NADE has a loose skeleton and an interconnected porous structure. With the increase in PTFE/CB mass ratio, more PTFE sinter appears on the catalytic layer. Consistent with the CA results, PTFE on the surface enhances the hydrophobicity of the catalytic layer and promotes the formation of a superhydrophobic interface. At the same time, it could be seen that serious agglomeration of PTFE occurred when PTFE/CB mass ratio reached 1.5.

**Electrocatalytic activity of NADE.** To further explore the structure and porosity of different PTFE-modified catalysts, the $N_2$ adsorption–desorption isotherms were studied. As shown in Fig. 2a, the isotherms were all type IV, confirming the coexistence of micropores, mesopores, and macropores[43,44]. As the amount of PTFE increased, the BET surface areas decreased from 171.34 to 43.85 m$^2$ g$^{-1}$, the volume of the micropores decreased, but the pore size increased (Supplementary Table 1). The contracting and agglomeration of PTFE during the sintering process are a likely reason for this phenomenon and it is consistent with the results in the literature[29]. Micropores provide active sites for ORR[45], but mesopores and macropores serve as gas transfer channels to supply oxygen[46]. As observed from Fig. 2a, micropores decreased when PTFE/CB increased from 0.1 to 1.5, but mesopores and macropores increased. When it was 0.6, it still had an obvious microporous structure and exhibited an advantage in the aperture range of 2–70 nm. These results indicated that the active sites for ORR decreased with the increase in PTFE content, but the oxygen mass transfer ability was enhanced. It is worth noting that not only the surface area and active sites, but also the efficient $O_2$ mass transfer are important for $H_2O_2$ production[46].

As mentioned above, we have obtained a stable superhydrophobic three-phase interface and the appropriate pore structure by adjusting the amount of PTFE in the catalytic layer, which have a

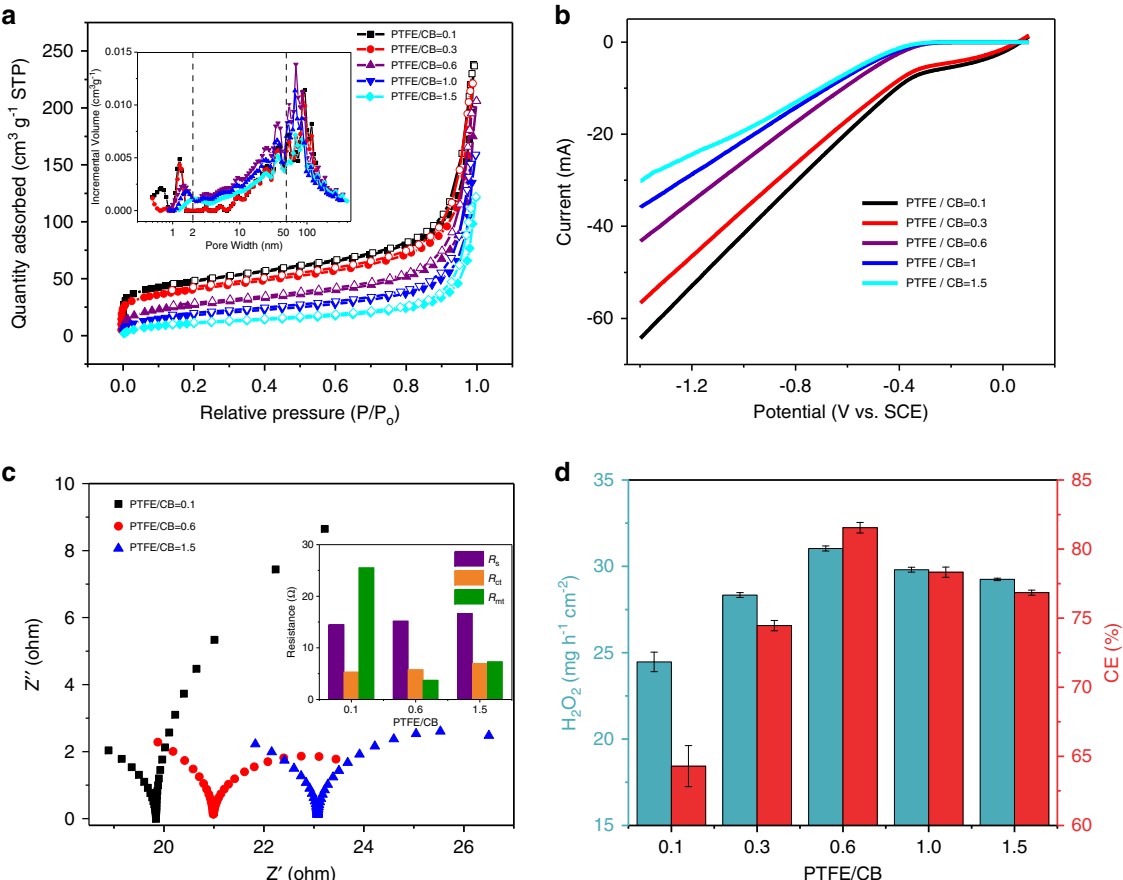

**Fig. 2 Characterization and H$_2$O$_2$ production of different NADEs. a** N$_2$ adsorption–desorption isotherms and pore-size distribution (inset) of catalytic layer materials with different PTFE/CB mass ratios. **b** LSV and **c** Nyquist plots/fitted curves of catalytic layer materials with different PTFE/CB mass ratios. **d** Performance comparison of different PTFE/CB NADEs for H$_2$O$_2$ generation and current efficiency at the current density of 60 mA cm$^{-2}$. The error bars represent the standard deviation of three independent samples. Source data are provided as a Source Data file.

great impact on the ORR activity and electrosynthesis of H$_2$O$_2$. To evaluate the ORR activity of the catalytic layer, the LSVs and Tafel plots of NADE with different PTFE/CB mass ratios were investigated (Fig. 2b; Supplementary Fig. 8). With the cathode potential scanning negatively, the current response was gradually increased. The highest PTFE content in the catalytic layer exhibited the lowest current response. As an important index to evaluate the ORR charge transfer[47,48], the exchange current density, $j_0$ is shown in Supplementary Fig. 9 and Supplementary Table 2, which fits linearly between 80 and 100 mV ($R^2 \geq 0.99$) according to the Tafel plots[49]. The results of $j_0$ are consistent with the analysis of LSVs, with values negatively related with PTFE content in the catalytic layer, which agrees well with Dong's works[50]. This is owing to the addition of PTFE reducing the reaction site, and excess non-conductive PTFE may hinder the transmission of electrons[29,30]. However, we have found that the existence of the hydrophobic interface is temporary when PTFE content is insufficient. With the increase in operation time, hydrophobicity decreases gradually; oxygen mass transfer will be limited, resulting in a decrease in catalytic efficiency. In Eq. 4, the electron transfer number ($n$) of each oxygen molecule in the ORR can be calculated. When the mass ratio of PTFE/CB is 0.6, $n$ was 2.07 (when $\beta$ is 0.07), indicating the best performance for H$_2$O$_2$ production.

To explore the interface properties of different NADEs, electrochemical impedance spectroscopy (EIS) was conducted. According to Nyquist plots and the fitted results (Fig. 2c), the solution resistance ($R_s$) and charge transfer resistance ($R_{ct}$) increased with the increase in PTFE content. The lowest $R_{ct}$ of the

NADE with PTFE/CB of 0.1 was ascribed to the excellent ion transfer between the hydrophilic electrode surface and electrolyte, which is consistent with the analysis above. However, NADE with PTFE/CB of 0.1 had the largest mass transfer resistance ($R_{mt}$) of 25.52 Ω compared with 3.73 Ω and 7.33 Ω for NADEs with PTFE/CB of 0.6 and 1.5. It indicated that the hydrophobic interface formed by the increase in PTFE content made oxygen supply sufficient, but excessive PTFE hindered ion transfer because the three-phase reaction sites diminished[51].

Under the combined effect of the factors mentioned above, the yields of H$_2$O$_2$ and current efficiency at different interfaces showed significant differences (Fig. 2d). The hydrophilic interface exhibited the lowest H$_2$O$_2$ production (24.86 mg h$^{-1}$ cm$^{-2}$) mainly due to the penetration of the catalytic layer by the electrolyte and insufficient oxygen transport channels. When the PTFE/CB mass ratio was 0.6, a stable superhydrophobic layer was formed, which could capture air in the atmosphere when in contact with liquid, thereby forming a gas, liquid, and solid coexisted interface at the micro-/nano level[52,53]. A moderate amount of PTFE promoted the formation of sufficient reaction sites in the catalytic layer and provides channels for rapid diffusion of oxygen, which would not seriously hinder the electron transfer[50]. As a result, the production of H$_2$O$_2$ (30.94 mg h$^{-1}$ cm$^{-2}$) and current efficiency (81.3%, 1 h) reached the highest value. Excessive PTFE did not increase the H$_2$O$_2$ yield, but caused it to decline because the number of micropores and specific surface area decreased sharply due to agglomeration of particles and sinter would hinder electron transfer. As illustrated in Supplementary Fig. 10, current efficiency

remained above 90% in the first 10 min, but gradually decreased with time. This was due to the decomposition of $H_2O_2$ caused by some parasitic reactions (e.g., spontaneous disproportionation of $H_2O_2$ to oxygen), and it would be exacerbated at high current densities. This was supported by the change of dissolved oxygen (DO) in two kinds of blank experiments and NADE system (Supplementary Fig. 11), in which the concentration of DO in NADE system continued to increase, while the blank experiments showed almost no change.

The loading of the catalyst layer has a great influence on $H_2O_2$ generation[5]. Figure 3 shows the yields of $H_2O_2$ under different loadings of the catalytic layer. It could be observed that the $H_2O_2$ production was the lowest when the loading was 4.4 mg cm$^{-2}$ (27.31 mg h$^{-1}$ cm$^{-2}$), and it was improved as the load increased, reaching the highest at 13.2 mg cm$^{-2}$ (30.93 mg h$^{-1}$ cm$^{-2}$). Accordingly, the current efficiencies of 4.4, 8.8, and 13.2 mg cm$^{-2}$ after running for 2 h were 63.9%, 66.8%, and 72.2%, respectively. With the catalyst loading increasing, the catalytic activity and electron transfer efficiency of NADE were enhanced. As demonstrated by LSV in Fig. 3c, the current response was greater at a higher catalyst loading ascribed to the increased catalyst mass loading. An electroactive surface area (ESA) describes the total surface that is in contact with the electrolyte and participated in the electron transfer process[54,55]. The ESA of NADEs with different catalyst loadings could be calculated based on the CV curve of $K_4[Fe(CN)_6]$ (Fig. 3d) and Randles–Sevcik equation (Eq. (5)). Consistent with the LSV results, the increases in loading that

promoted an increase in ESA, from 4.4 to 13.2 mg cm$^{-2}$, were 1.86, 4.47, and 5.62 cm$^2$, respectively. On the other hand, when the catalyst loading was too high, the thickness of the catalytic layer increased, and the oxygen diffusion capacity within the electrode was limited, resulting in the effect on ORR performance[27,56]. Pérez et al. reported that the increasing loading of the catalyst caused an important diffusion resistance and a sharp decrease in the oxygen supplied to the reaction interface, slowing down the reaction rate of $H_2O_2$ electrosynthesis[40]. When the catalyst loading was as low as 1–2 mg cm$^{-2}$, although the current efficiency was slightly reduced, NADE also had excellent $H_2O_2$ production efficiency with the current efficiency of about 80% at 10 min (Supplementary Fig. 12). NADE maintained stable $H_2O_2$ production efficiency (CE = 63.8–78.6%; 1 h) after running 10 times (e.g., 20 h) at the current density of 60 mA cm$^{-2}$ with the catalyst loading of 1 mg cm$^{-2}$ (Supplementary Fig. 13). However, NADE with low catalyst loading had some drawbacks when it was operated for a long time or at high current density. As shown in Supplementary Fig. 14a and b, after 20 h of operation, NADE with the catalyst loading of 1 mg cm$^{-2}$ showed significant electrolyte penetration compared with NADE with the catalyst loading of 13.2 mg cm$^{-2}$. When the applied current density reached 200 mA cm$^{-2}$, NADE with catalyst loadings of 2 and 4.4 mg cm$^{-2}$ showed different degrees of electrolyte penetration, while NADE with catalyst loading of 8.8 mg cm$^{-2}$ did not occur at all (Supplementary Fig. 14c–e). Therefore, it is also necessary to control the catalyst loading within an appropriate range to achieve the best performance for NADE.

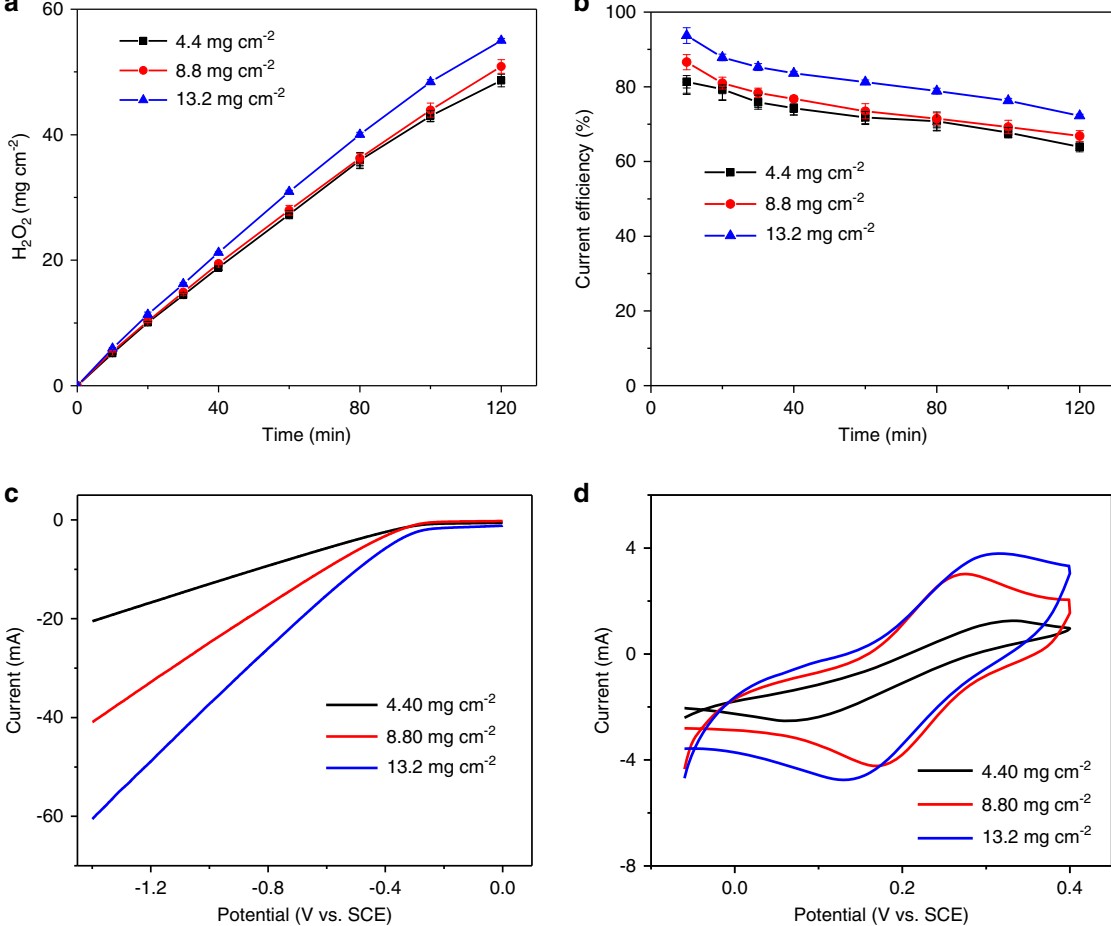

**Fig. 3 Effect of catalyst loading on the ORR performance.** The effect of catalyst loading on **a** the yields of $H_2O_2$, **b** current efficiency, **c** LSV, and **d** CV in 1 M KCl containing 10 mM $K_4[Fe(CN)_6]$. The error bars in **a**, **b** represent the standard deviation of three independent samples. Source data are provided as a Source Data file.

**Oxygen mass transfer and electron transport in the ORR**. In order to verify the efficient oxygen transfer performance of NADE, an experiment without air/$O_2$ pumping was conducted, in which the cathode was immersed in the solution and connected to the air through a superhydrophobic substrate (Fig. 4a (ii)). In this case, the electrode was surrounded by the solution and oxygen could only diffuse through a narrow section. Similarly, a diphase system that only trapped DO in solution was also

compared (Fig. 4a (iii)), showing a very small $H_2O_2$ accumulation. As a result, NADE system with one side connecting the atmosphere exhibited the best $H_2O_2$ electrosynthesis performance and the highest current efficiency under the same conditions as the other two cases (Fig. 4a (i) and Supplementary Fig. 15). This was mainly due to the fact that other two cases were limited by oxygen mass transfer, resulting in a decrease in ORR activity. The oxygen reduction current on NADE did not change when

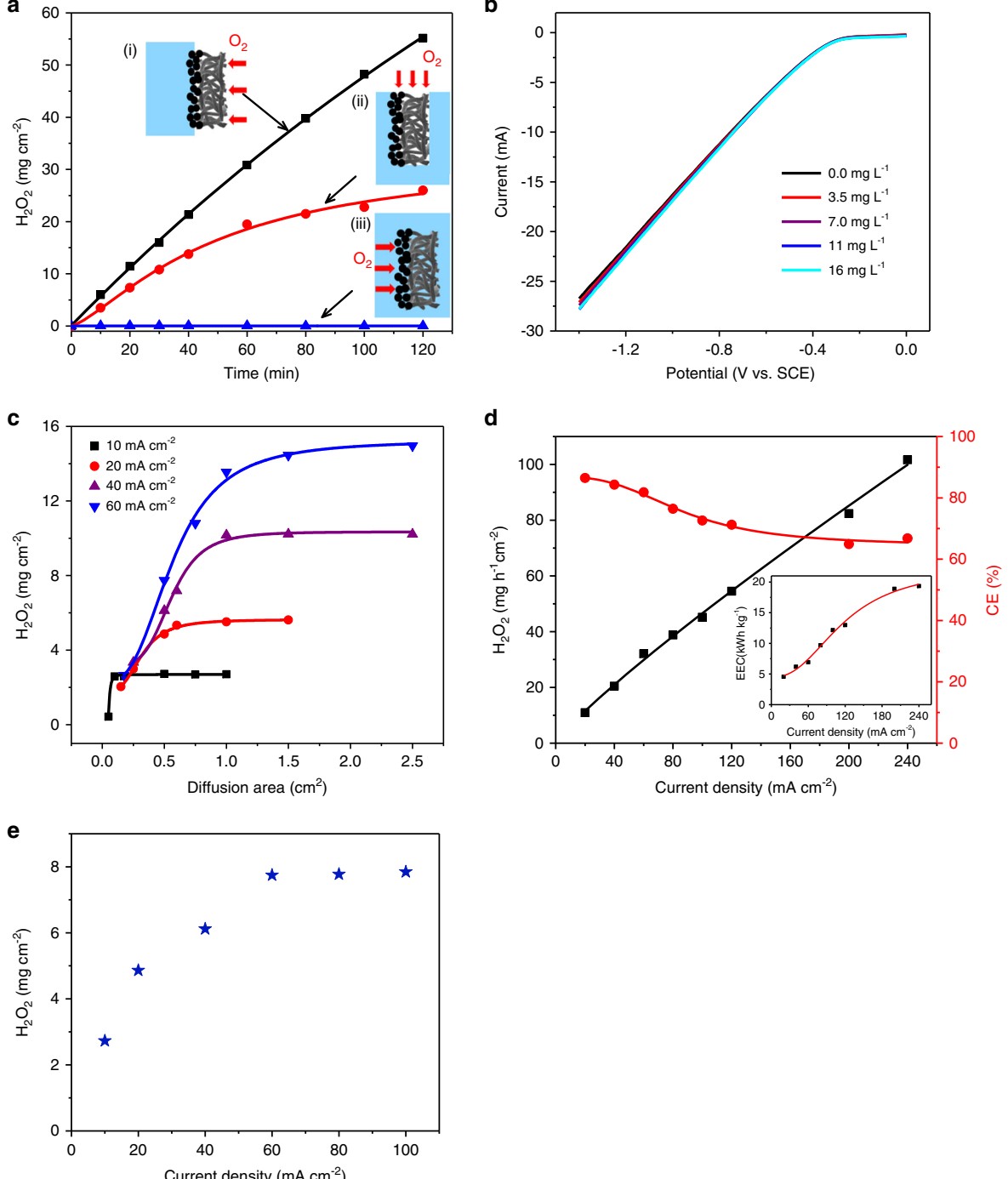

**Fig. 4 Oxygen mass transfer and electron transport in the ORR. a** The concentration of $H_2O_2$ production by different systems at 60 mA cm$^{-2}$ current density. **b** LSVs of NADE in Na$_2$SO$_4$ solution with different oxygen concentrations. **c** The concentration of $H_2O_2$ production (0.5 h) with the change in oxygen diffusion area under different current densities. **d** The yield of $H_2O_2$ and the corresponding current efficiency under different current densities; the inset shows the corresponding energy consumption at different current densities. **e** Production of $H_2O_2$ under different current densities with the diffusion area of 0.5 cm$^2$. Source data are provided as a Source Data file.

operated in electrolytes with different oxygen concentrations, indicating that the oxygen mass transfer in ORR was determined by gas-phase oxygen diffusion and no longer affected by fluctuations of DO in the solution (Fig. 4b). When the CF substrate without loading the catalytic layer was used for $H_2O_2$ electrosynthesis, very little $H_2O_2$ of $0.92\,mg\,cm^{-2}\,h^{-1}$ was produced (Supplementary Fig. 16). Compared with the total $H_2O_2$ production of $30.94\,mg\,h^{-1}\,cm^{-2}$, it proved that the contribution of the CF substrate in the NADE system to the total $H_2O_2$ production was very small.

To reveal the $O_2$ mass transfer and electron transport in the ORR, and demonstrate the contribution in improving oxygen utilization in the NADE system, we conducted comparative experiments to control the oxygen supply by changing the diffusion area under different current densities. As mentioned above, the oxygen supply for NADE depends on actively diffused oxygen through the superhydrophobic CF matrix. Assuming that the oxygen diffusion mode is one-dimensional diffusion and macroscopically follows the first law of Fick gas diffusion[57], the oxygen supply can be controlled by changing the diffusion area. As presented in Fig. 4c and Supplementary Fig. 17, when the diffusion area was small ($0.175\,cm^2$), oxygen naturally diffusing to the reaction interface was very less, resulting in the ORR activity being limited by oxygen mass transfer, and the production of $H_2O_2$ was very low even if the current density was high ($60\,mA\,cm^{-2}$). As the diffusion area expanded, the amount of oxygen supplied to the reaction interface gradually increased, resulting in a rapid increase in $H_2O_2$ concentration. When the diffusion area reached $2.5\,cm^2$, the amount of oxygen was enough, as it tended to be stable and changed into an electron transfer control process due to the limited current applied. At other current densities (10, 20, and $40\,mA\,cm^{-2}$), the same oxygen mass transfer limitation was observed.

One of the industrial requirements of a suitable cathode for $H_2O_2$ production is that it can work at high current density[58]. Barros et al. operated a normal GDE at the current density of $212\,mA\,cm^{-2}$; though the $H_2O_2$ production could reach $44.9\,mg\,h^{-1}\,cm^{-2}$, the current efficiency was as low as 33.3%[59]. We subsequently increased the current density and observed that the production of $H_2O_2$ increased linearly with its increase ($R^2 = 0.995$) within the scope of the investigation. Even if the current density was increased to $240\,mA\,cm^{-2}$, ORR was still not limited by oxygen mass transfer, and the $H_2O_2$ production reached $101.67\,mg\,h^{-1}\,cm^{-2}$ (Fig. 4d; Supplementary Fig 17e). Although increasing the current density led to a decrease in current efficiency (from 86.5% to 66.8%) and an increase in energy consumption (from 4.6 to $19.4\,kWh\,kg_{H2O2}^{-1}$), the performance would still be satisfactory, breaking the limitation that existed in normal GDE that operation at high current density would cause extremely low current efficiency (Supplementary Table 3). This point will be very vital and favorable when electrochemical devices are scaled up from laboratory scale to industrial applications[5]. Therefore, NADE with simple preparation and cheap material has excellent in situ $H_2O_2$ production performance, which is beneficial to the application of Fenton-based EAOPs[60].

The oxygen mass transfer coefficient of NADE diffusion layer calculated above ($1.15 \times 10^{-1}\,cm^2\,s^{-1}$) was verified by experiments. We explored the relationship of the maximum $H_2O_2$ production of NADE with current density at the same diffusion area of $0.5\,cm^2$ (Fig. 4e). It increased linearly with increasing current density, but basically remained unchanged at the current density of $60\,mA\,cm^{-2}$, indicating that $H_2O_2$ production turned into oxygen mass transfer controlled when the current density was higher than this point[10]. Assuming the OUE of 100%, the minimum $O_2$ diffusion coefficient of the diffusion layer was predicted to be $0.75 \times 10^{-1}\,cm^2\,s^{-1}$, meaning the actual $O_2$ diffusion coefficient of the porous diffusion layer should be between $0.75 \times 10^{-1}$ and $2 \times 10^{-1}\,cm^2\,s^{-1}$ (the $O_2$ diffusion coefficient in air[35]) and agreed with our calculation.

**Performance comparison with normal GDE and stability test.** Based on these results above, the NADE enables efficient synthesis of $H_2O_2$ with its excellent air diffusion performance without external air pumps, which may overturn normal GDE and provide a new approach for on-site production of $H_2O_2$. Here, we compared the $H_2O_2$ production of NADE with the conventional GDE prepared by the rolling method[61]. It is worth noting that normal GDE required an external air pump to provide a gas supply of 500 mL $min^{-1}$ (5 W), while NADE relied on oxygen that actively diffuses from the atmosphere. Interestingly, as shown in Fig. 5a, the yield of $H_2O_2$ on NADE at $20\,mA\,cm^{-2}$ was 21.7% higher than that of the normal GDE (10.97 and $9.01\,mg\,h^{-1}\,cm^{-2}$, respectively), but at $200\,mA\,cm^{-2}$, it was 61.2% higher (83.37 and $51.73\,mg\,h^{-1}\,cm^{-2}$, respectively). In the comparison of current efficiency, the normal GDE at $200\,mA\,cm^{-2}$ was 40.78%, while the NADE corresponded to 65.72% (Fig. 5b). The performance of normal GDE was consistent with the results in literature[23,40,59]. With the increase in current density, more $O_2$ was exhausted by the ORR, which led to the $O_2$ transport limitation[62]. On the other hand, parasitic reactions enhanced, resulting in the $H_2O_2$ production that increased slowly. However, NADE showed better $H_2O_2$ production and current efficiency at high current density. It may be attributed to the excellent oxygen mass transfer efficiency. It is worth noting that the cumulative $H_2O_2$ of normal GDE in 2 h without the air pump was only $8.15\,mg\,h^{-1}\,cm^{-2}$ at most (Supplementary Fig. 18). In the absence of additional air-pumping equipment, the production of $H_2O_2$ decreased as the current density increased due to the seepage on normal GDE and parallel reactions resulting in decomposition of $H_2O_2$[23].

Compared with other literatures, the NADE system proposed in this work had an obvious advantage in $H_2O_2$ yield, current efficiency, and OUE. As presented in Fig. 5c-d and Supplementary Table 4, the comparison suggested that the performance of NADE (marked as red star) is, to a large extent, superior to all other air-breathing electrodes (marked as square, see refs. [63,64]) and normal GDE supplemented by aeration or pressurized oxygen (marked as triangle, see refs. [27–29,31,39,65–69]). The $H_2O_2$ yield of other electrodes was almost below $30\,mg\,h^{-1}\,cm^{-2}$, and the energy consumption ranged between 15.9 and $53.9\,kWh\,kg^{-1}$. In contrast, when the $H_2O_2$ production rates of this work were 10.97, 30.93, and $101.67\,mg\,h^{-1}\,cm^{-2}$, the corresponding energy consumption was only 4.55, 6.92, and $19.35\,kWh\,kg^{-1}$, and OUE (44.5–64.9%) was much higher than the values in other literature as far as we know. Furthermore, GDE's aeration energy consumption was yet not low ($0.04$–$0.5\,kWh\,m^{-3}$)[70], but it is generally ignored. This work will completely reduce this part of energy consumption, which is more conducive to the cost-effective production of $H_2O_2$.

To inspect the stability of NADE, 10-times consecutive tests of $H_2O_2$ production were conducted. As presented in Fig. 5d, the $H_2O_2$ yield and current efficiency were very stable with a maximum 5% decrease during the 10-time (20 h) runs at $60\,mA\,cm^{-2}$, suggesting the possibility of NADE application for extended periods.

To further explore the potential of the NADE application for wastewater treatment, several typical pollutants were selected as model organic contaminants in EF and PEF systems. The removal efficiency of pollutants and TOC is presented in Supplementary Figs. 19–21. Although the initial concentration of pollutants was high, all the pollutants' removal efficiencies could achieve 100% by EF process within 90 min, especially, SMT, MB, and TC were

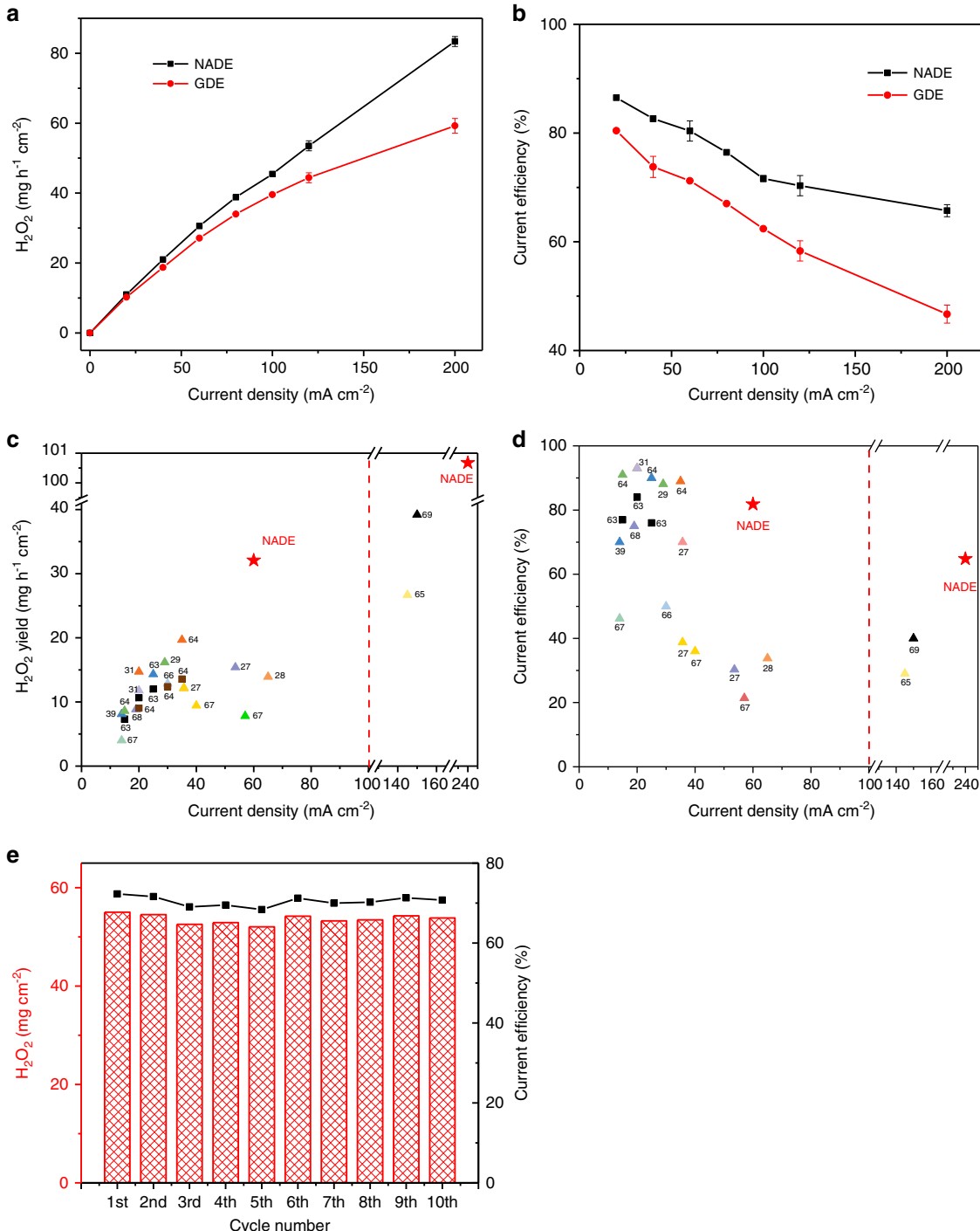

**Fig. 5 Performance comparison with normal GDE and stability test. a** Concentrations of $H_2O_2$ and **b** current efficiency comparisons of the NADE system and passive diffusion GDE (supply air at a flow rate of 0.5 L min$^{-1}$) at different current densities. Comparison of **c** $H_2O_2$ yield and **d** current efficiency with the literature. **e** The stability test of the NADE system in 10-times (20 h) continuous runs. The error bars in **a**, **b** represent the standard deviation of three independent samples. Source data are provided as a Source Data file.

removed completely within 60 min (Supplementary Fig. 19a). The TOC removal of 2,4-D, phenol, MB, TC, and SMT at 2 h were 51.1%, 55.1%, 62%, 57.5%, and 39.5%, respectively. The corresponding MCE were 35.3%, 79.7%, 45.7%, 49%, and 28.4%. The comparison with literature on the removal efficiency and energy consumption is listed in Supplementary Table 5. García et al. degraded 2,4-D and reported a 59% removal of TOC with MCE of 23% at 300 min. However, the energy consumption was 420 kWh kgTOC$^{-1}$, which was much higher than that of our work

(66.4 kWh kgTOC$^{-1}$)[71]. Compared with the TOC removal efficiency and energy consumption of these pollutants in literature, EF process based on NADE had a good performance, especially lower energy consumption (36.6–91.7 kWh kgTOC$^{-1}$)[22,72,73].

PEF process has faster degradation and mineralization efficiency of pollutants than EF process benefiting from rapid reduction of ferric iron and direct photocatalysis of $H_2O_2$[74]. For example, all investigated contaminants could be completely removed within 30 min, except MB (96.8%) and phenol

(92.1%). The removal of TOC reached 86.6% (2,4-D), 89.2% (phenol), 74.3% (MB), 81.3% (TC), and 63.8% (SMT) within 2 h, respectively. For the mineralization of phenol, Yue et al. and Assumpcao et al. reached 85% and 11% in 3 h and 2 h, respectively[20,75]. It is worth mentioning that all experiments in the literature need additional aeration, but this experiment did not, which means that a large part of energy consumption is saved. All the results demonstrate that NADE is a promising cathode used in $H_2O_2$-based EAOPs for wastewater treatment.

## Discussion

In summary, we have developed a superhydrophobic NADE system based on natural air diffusion, which greatly improves $H_2O_2$ production without external oxygen-pumping equipment. First, the higher oxygen mass transfer coefficient and air−liquid −solid reaction interface solve the oxygen mass transfer limitation in ORR, even at high current densities, resulting in a significant increase in $H_2O_2$ production (101.67 mg h$^{-1}$ cm$^{-2}$), current efficiency (66.79%), and OUE (44.5–64.9%). Second, the superhydrophobic NADE system brings an exploratory platform for understanding and minimizing oxygen mass transfer restriction in ORR, and highlights the importance of the design of hydrophobicity/hydrophilicity reaction interface for $H_2O_2$ electrosynthesis. NADE system has excellent ability in electrosynthesis of $H_2O_2$, which makes it possible to remove pollutants rapidly and thoroughly even at high concentration. Significantly, this superhydrophobic NADE system is stable and cost-effective without additional aeration energy consumption; thus, it would have great potential to replace normal GDE for $H_2O_2$ production, which is presently widely used in EAOPs.

## Methods

**Chemicals and materials**. All chemicals used in this study were analytical grade. 2,4-dichlorophenoxyacetic acid (2,4-D) and phenol were purchased from Beijing Lideshi chemical technology Co., Ltd. Methylene blue (MB), sulfamethazine (SMT), and tetracycline (TC) were purchased from Aladdin (Shanghai). The CF (Beijing Jinglong), PTFE, and CB (Shanghai Hesen) were used for NADE preparation.

**Fabrication of NADE**. The schematic diagram of NADE manufacturing process is shown in Fig. 1a, where electrocatalysts were immobilized on one side of a modified CF (2.5-mm thick). After ultrasonically cleaning in deionized water and ethanol in sequence for 15 min, the cleaned CF was soaked in PTFE suspension (2 wt%) for 10 min, then taken out to dry, and calcined at 360 °C for 30 min. The mixture of CB, absolute ethyl alcohol, and PTFE suspension with five PTFE/CB mass ratios of 0.1, 0.3, 0.6, 1, and 1.5 were coated as the catalyst layer. Finally, the electrodes loaded with the catalytic layer were calcined at 360 °C for 30 min.

**Characterization and evaluation of NADE**. The surface morphology of NADE was measured with field-emission scanning electron microscope (LEO-1530VP). The contact angle measurements of modified CF and catalytic layers with different PTFE/CB mass ratios were performed using a goniometer (JC2000D1, POWER-EACH). Linear-sweep voltammetry was performed on CHI660D workstation at a scan rate of 10 mV s$^{-1}$ in a three-electrode system, in which NADE was the working electrode, DSA was the counter electrode, and saturated calomel electrode was the reference electrode. The specific surface area and pore-size distributions were collected by $N_2$ adsorption–desorption isotherms using Brunaure–Emmett–Teller and Barrett–Joyner–Halenda methods (ASAP 2460).

**Electrochemical experiments**. Electrocatalytic $H_2O._2$ production was performed in a 250-mL reactor, using NADE (3 cm × 1.7 cm) as cathode and DSA (2 cm × 4 cm; $IrO_2$ coating) as anode (Fig. 1d). $Na_2SO_4$ (0.05 M) was used as the supporting electrolyte.

Direct electrosynthesis of $H_2O_2$ can be applied to environmental remediation through various processes; EF and PEF experiments were carried out for degrading different types of pollutants with 100 mg L$^{-1}$ concentration, including 2,4-D, phenol, MB, TC, and SMT. The reactor and electrodes used in the degradation experiments were the same as those used in the above experiments. In PEF treatments, the solution was irradiated with a UVC lamp ($\lambda_{max}$ = 254 nm; 5 W) placed on top of the electrochemical reactor. The solution pH was adjusted to 3 with $H_2SO_4$ and the added ferrous sulfate was 0.5 mM.

**Analytic methods**. The concentration of $H_2O_2$ was measured using the method of potassium titanium (IV) oxalate by UV–Vis spectrophotometer at a wavelength of 400 nm. The current efficiency (CE, %) for $H_2O_2$ synthesis was calculated as Eq. (1)

$$\text{CE} = \frac{2CVF}{Q} \qquad (1)$$

where $C$ is the concentration of $H_2O_2$ (mol L$^{-1}$), $V$ is the volume of solution (L), $F$ is the Faraday constant (C mol$^{-1}$), and $Q$ is the amount of charge passed through the cathode (C).

The oxygen utilization efficiency (OUE, %) describing the ratio of $O_2$ that was used for $H_2O_2$ synthesis, was calculated by Eq. (2)

$$\text{OUE} = \frac{D \cdot A}{M_{O_2} \cdot n_{H_2O_2}} \cdot \frac{d\rho}{dx} \times 100\% \qquad (2)$$

where $D$ is the diffusion coefficient of oxygen (cm$^2$ s$^{-1}$), $A$ is the diffusion area (cm$^2$), $\rho$ is the oxygen concentration (g cm$^{-3}$), $M_{O_2}$ is the molar mass of oxygen (g mol$^{-1}$), $n_{H_2O_2}$ is the molar amount of the generated $H_2O_2$ (mol), and $x$ is oxygen diffusion distance (cm).

The electric energy consumption (EEC, kWh kg$^{-1}$) was calculated by Eq. (3)[74]

$$\text{EEC} = \frac{1000UIt}{CV} \qquad (3)$$

where $U$ and $I$ are the applied voltage (V) and current (A), respectively, $t$ is the electrolysis time (h), $C$ is the concentration of $H_2O_2$ (mg L$^{-1}$), and $V$ is the volume of solution (L).

The electron transfer number $n$ during the ORR can be estimated from the slopes of the linear region in the Tafel plot via the following Eq. (4)

$$\lg|j| = \lg j_0 + (\beta nF/2.303RT)|\eta| \qquad (4)$$

where the slope is $\beta F/2.303RT$ and the symmetry factor $\beta$ is related to pH (usually < 0.5 in the cathodic ORR).

Electroactive surface area (ESA) was measured by analyzing the CV curves with Randles–Sevcik Eq. (5)[76]

$$I_P = 2.69 \times 10^5 \cdot n^{3/2} \cdot A \cdot D^{1/2} \cdot C \cdot \nu^{1/2} \qquad (5)$$

where $I_P$ is the peak current (A), $n$ is the number of electrons contributing to the redox reaction, $D$ is the diffusion coefficient ($6.3 \times 10^{-6}$ cm$^2$ s$^{-1}$), $C$ is the concentration of Fe(CN)$_6^{4-}$ in the bulk solution (mol cm$^{-3}$), $\nu$ is the scan rate (V s$^{-1}$), and $A$ is the ESA (cm$^2$).

The concentration of 2,4-D, phenol, TC, and SMT was analyzed by high-performance liquid chromatography (U3000, ThermoFisher) with a DAD detector. The separations were performed on Acclaim™ 120 C18 column (3 μm, φ3.0 × 100 mm) using the mobile phase of 60:38 (v/v) methanol/2% acetic acid at a flow rate of 0.3 mL min$^{-1}$. The mobile phase for TC and SMT was 60:40 (v/v) methanol and water, operated at a flow rate of 0.1 mL min$^{-1}$ with Acquity UPLC BEH C18 column (1.7 μm, φ2.1 × 100 mm). The concentration of MB was measured by the UV − vis spectrophotometer (UV-2600, Shimadzu) at a wavelength of 664 nm. Total organic carbon (TOC) was measured by TOC-L (Shimadzu).

The removal efficiency of pollutant ($\eta$) was calculated according to Eq. (6)

$$\eta = \frac{C_0 - C_t}{C_0} \times 100\% \qquad (6)$$

where $C_0$ and $C_t$ (mg L$^{-1}$) are the initial and final concentration of pollutants.

TOC removal ($\theta$) was calculated by Eq. (7)

$$\theta = \frac{\text{TOC}_0 - \text{TOC}_t}{\text{TOC}_0} \times 100\% \qquad (7)$$

where $\text{TOC}_0$ and $\text{TOC}_t$ (mg L$^{-1}$) are the initial and final TOC of pollutants.

Mineralization current efficiency (MCE) was calculated by Eq. (8)

$$\text{MCE} = \frac{nFV\Delta\text{TOC}}{4.32 \times 10^7 mIt} \times 100\% \qquad (8)$$

where $n$ is the number of electrons consumed for complete mineralization of each pollutant molecule, $F$ is the Faraday constant (C mol$^{-1}$), $V$ is the volume of solution (L), $4.32 \times 10^7$ is a conversion factor, $m$ is the carbon atom number of each pollutant molecule, $I$ is the applied current (A), and $t$ is the electrolysis time (h).

**Development of the oxygen diffusion coefficient model**. A specified expression of the oxygen diffusion coefficient under dry condition is proposed in Eqs. (9–12)[77], where Fick diffusion, Knudsen diffusion, and transition diffusion are concerned, and the three diffusion modes are quantified by the aperture-density function

$$D_{fe} = \frac{\emptyset}{\sqrt{3} \cdot \tau} \cdot \left( \sum_{i=1}^{x} D_{K,i} \cdot v_i + \sum_{j=1}^{y} D_{C,j} \cdot v_j + D_{F,f} \cdot v_f \right) \qquad (9)$$

$$D_K = \frac{1}{3} \cdot \varphi \cdot \sqrt{\frac{8RT}{\pi M}} \qquad (10)$$

$$D_F = \frac{1}{3} \cdot \bar{\lambda} \cdot \sqrt{\frac{8RT}{\pi M}} \qquad (11)$$

$$\frac{1}{D_C} = \frac{1}{D_F} + \frac{1}{D_K} \qquad (12)$$

where $D_{fe}$ (m$^2$ s$^{-1}$) is the effective diffusion coefficient of O$_2$, $\Phi$ is the porosity of porous matrix, $\tau$ is the average tortuosity of the diffusion layer matrix, $\lambda$ is the oxygen mean-free path (m), $\varphi$ is the effective aperture (m), $D_{K,i}$ (m$^2$ s$^{-1}$) is the Knudsen diffusion coefficient in parallel tube bundles with an effective pore diameter of $\varphi_i$ ($0 < \varphi_i < 0.1\lambda$), $D_{C,j}$ (m$^2$ s$^{-1}$) is the transition diffusion coefficient in parallel tube bundles with an effective pore diameter of $\varphi_j$ ($0.1\lambda < \varphi_j < 10\lambda$), $D_{F,f}$ (m$^2$ s$^{-1}$) is the Fick diffusion coefficient in parallel tube bundles with an effective pore diameter of $\varphi_f$ ($\varphi_f > 10\lambda$), $v_f$ is the proportion of pores with an effective pore diameter greater than ten times the mean-free path of O$_2$ molecules, and $v_i$ and $v_j$ are the proportions of holes with effective apertures $\varphi_i$ and $\varphi_j$, respectively.

## Data availability

The source data underlying Figs. 2b–d, 3, 4, 5a–b, 5e and Supplementary Figs. 8, 10–13 and 15–21 are provided as a Source Data file. Extra data are available from the corresponding author upon reasonable request.

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

## Acknowledgements

This work was supported by the National Natural Science Foundation of China (nos. 21773129, 21811530274, and 21976096), Tianjin Science and Technology Program (19PTZWHZ00050), Tianjin Development Program for Innovation and Entrepreneurship, National Key Research and Development Program of China (2016YFC0400706), National Special S&T Project on Water Pollution Control and Management (2017ZX07107002), 111 program, Ministry of Education, China (T2017002), and Fundamental Research Funds for the Central Universities.

## Author contributions

Q.Z. and M.Z. conceived the idea. Q.Z. conducted the experiments. G.R. and Y.L. (Yawei Li) provided constructive suggestion and discussion. Y.L. (Yanchun Li) helped the preparation of NADE. X.D. helped on analysis of electrochemical test results. Q.Z. and M.Z. analyzed the data and cowrote the paper with discussion with all authors.

## Competing interests

The authors declare no competing interests.

## Additional information

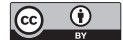

