## [Peer Review File · Nature Communications]

Reviewers' Comments:

Reviewer #1:

Remarks to the Author:

This manuscript reports the preparation of a so-called "superhydrophobic natural air diffusion electrode (NADE)" to improve oxygen diffusion, and then to enhance the H₂O₂ generation performance from ORR reaction. This NADE was simply prepared by coating the carbon felt electrode with optimized hydrophobic PTFE. The author emphasized in this manuscript that the oxygen utilization efficiency was significantly improved by PTFE modification. The catalyst coated NADE showed a H₂O₂ FE of around 80% with a production rate of ~80 mg h⁻¹ cm⁻². Using PTFE to make a superhydrophobic electrode for oxygen reduction was a widely reported strategy. The 2e-ORR performance is also not impressive compared to current literature. Thus, I do not recommend the publication of this paper.

Here are some minor comments for authors' reference when they prepare their work for other more specialized journals:

- 1) A very high loading of catalyst (13.2 mg cm⁻²) was used to to achieve the highest current efficiency in this work, which is much higher than the typically loading. Can the author explain why such a high loading was needed to achieve a relative low H₂O₂ FE of 72.7%?
- 2) "As verified by LSV in Fig. 3c, a higher current response was observed upon a higher catalyst loading, means faster electron transfer in the circuit." I guess the higher current response for higher catalyst loading sample was ascribed to the increased mass loading. I do not think this indicates the faster electron transfer. Can the author provide the mass loading normalized LSV curve?
- 3) At line 205, the author claimed that when the mass ratio of PTFE/CB is 0.6, the electron transfer number was calculated as 2.07, indicating an exclusive 2e-ORR process. However, the reported maximal H₂O₂ FE of the PTFE/CB-0.6 sample was around 80%. Can the author explain why?

Reviewer #2:

Remarks to the Author:

In this paper, a new superhydrophobic carbon felt gas diffusion layer was constructed to greatly improve oxygen mass transfer efficiency. Interestingly, the natural air diffusion electrode has an oxygen utilization rate of up to 44.5%–64.9% without external aeration. In a word, the manuscript is of quality if the following comments could be addressed properly:

1. P 6, L 7, the general gas diffusion electrodes are prepared by rolling method, and the catalytic layer and diffusion layer can be tightly combined. For this NADE, does the author evaluate the adhesion strength of catalytic layer on carbon felt substrate? Could you please give a brief explanation about using carbon felt as the base for gas diffusion electrode? Can other porous materials be used for the substrate of NADE?
2. P 7, L 15-17, as the author mentioned, the three-phase interfaces on catalytic layer had a short life and were easily flooded in electrolyte, an important reason is that according to the electrowetting theory, compared with the non-applied voltage, the contact angle between the solid and liquid would decrease when an external voltage was applied. How do you conclude that "after the ratio was increased to 0.6, a stable superhydrophobic interface was formed and the three-phase interface basically didn't change with the time"? Does the author check the contact angle change of the electrode after a period of electrolysis?
3. P 12, L 13-14, will the carbon felt substrate in the NADE system undergo 2-electron pathway ORR to generate H₂O₂? Does this contribute to the total H₂O₂ production of the electrode?
4. P 12, L16, how do you explain the trend of H₂O₂ accumulation in Fig. 4a-C, why H₂O₂ does not seem to accumulate under this condition? Dissolved oxygen can also participate in ORR to generate

H₂O₂ in situ.

5. P20, L 17-22, please check the layout of equations 6-9 again.

6. In the supplementary materials, please check the number of references. Cited papers after [19] were not found in the text.

7. In Fig.1, the oxygen transports only through the GDE, I consider the transport pathway from the air/liquid interface maybe more important since the resistance is lower.

Reviewer #3:

Remarks to the Author:

The authors present a novel natural air diffusion electrode (NADE) that shows great promise for upscaling the electrochemical production of H₂O₂ via the two-electron oxygen reduction reaction (ORR). This topic is well motivated and timely due to the imminent need for a sustainable and decentralised alternative to the energy-intensive anthraquinone process. Experimental testing of the superior NADE performance is thorough, well-argued and overall convincing. As such, I recommend this article for publication in Nature Communications after the authors have addressed the minor revisions noted below:

1. Did the authors measure product distributions in ORR with the NADE vs the GDE? Are the <100% current efficiencies due to the competing 4e- ORR toward H₂O, or something else? I think that including this information would very much help improve this paper.

2. The manuscript must be proofread carefully to correct for grammatical errors that currently make some parts of the text difficult to understand. There are quite a few such instances throughout the manuscript. A few examples are listed below:

Lines #199-201: " This is owing to the addition of PTFE reduces the reaction site and PTFE is not electrically conductive so that excess may hinder the transmission of electrons."

Lines #240-241: " Electroactive Surface Area (ESA), describing the total surface which was accessible to the electrolyte and could participate in the electron transfer process."

Lines #276-278: "As presented in Fig. 4c and Supplementary Fig. 10, when the diffusion area was small (0.175 cm²), the amount of oxygen diffused to the reaction interface by the concentration difference was very few..."

3. I am a bit confused about the optimum PTFE/CB ratio reported by Yu et al. (Ref. 32) as compared to the one reported in this work. If I understand correctly, Yu et al. reported an optimum PTFE/CB=5 (in fact I suggest that the authors adopt this annotation also on line #96 so that it is consistent with the presentation of their results in Figure 2). However, this work reports an optimum PTFE/CB=0.6 and suggest that higher PTFE/CB ratios result in worse performance due to particle agglomeration and sintering. Can the authors explain and comment on this difference?

4. On line #62: I think the authors should be writing "As the applied potential increases..." rather than "As the current density increases..."

5. On line #197 the authors write "... referring to the report". What report is this?

6. The abbreviation 'EIS' on line #208 needs to be explained.

Response to the reviewers' comments

Ms. Ref. No.: NCOMMS-19-38589

Title: Ultra-efficient electrosynthesis of hydrogen peroxide on superhydrophobic three-phase interface by natural air diffusion

Reviewers' comments:

Reviewer #1:

This manuscript reports the preparation of a so-called “superhydrophobic natural air diffusion electrode (NADE)” to improve oxygen diffusion, and then to enhance the H₂O₂ generation performance from ORR reaction. This NADE was simply prepared by coating the carbon felt electrode with optimized hydrophobic PTFE. The author emphasized in this manuscript that the oxygen utilization efficiency was significantly improved by PTFE modification. The catalyst coated NADE showed a H₂O₂ FE of around 80% with a production rate of ~80 mg h⁻¹ cm⁻². Using PTFE to make a superhydrophobic electrode for oxygen reduction was a widely reported strategy. The 2e-ORR performance is also not impressive compared to current literature. Thus, I do not recommend the publication of this paper.

Response: We sincerely thanked the reviewer's comments.

The key advance of this work was that we proposed a novel natural air diffusion electrode (NADE) to greatly improve the production of H₂O₂ from ORR reaction without artificial supply of oxygen. This NADE reduced the

equipment investment and operating power consumption required for aeration, which greatly reduced the energy consumption of electrosynthesis of H₂O₂. At the same time, the NADE also overcame the limitation of high current density on the electrosynthesis of H₂O₂ in the ORR process, and achieved the H₂O₂ production of 100.67 mg h⁻¹ cm⁻² at current density of 240 mA cm⁻², which would be very vital and favorable when scaling up electrochemical cells from the laboratory scale to the industrial scale application. Under similar experimental conditions, the H₂O₂ production, current efficiency and total energy consumption of NADE were superior to all other air breathing electrodes and normal GDE supplemented by aeration or pressurized oxygen, especially at high current densities (as we shown in Fig. 5c-d and Supplementary Table 4).

As you said, the use of PTFE to prepare superhydrophobic electrodes for oxygen reduction was a widely reported strategy because it is a simple and feasible way. Similarly, the catalyst we used for 2e⁻-ORR was ordinary carbon black (CB) that could be directly obtained from the market at low cost (¥320 / 1000 g) because we think simple and economical electrodes are more likely to upscale the electrochemical production of H₂O₂ via the 2e⁻-ORR. However, NADE is different from the electrodes reported in the literature. For the first time, we proposed this unique electrode using a superhydrophobic carbon felt as a diffusion layer and loading a suitable ratio of PTFE/CB as a catalytic layer, which can achieve the above-mentioned excellent performance driven by a novel natural air diffusion in electrosynthesis of H₂O₂. Since these experiments were

operated in the undivided electrochemical device without ion exchange membrane, H_2O_2 formed at the cathode can be degraded at the anode. Other deleterious reactions including spontaneous disproportionation of H_2O_2 to give H_2O and $\frac{1}{2}\text{O}_2$ and H_2O_2 can undergo (deleterious) further oxidation or reduction at the electrode due to the insufficient mass transfer of H_2O_2 in the liquid phase also cause current efficiency to decrease over time, especially at high current densities.

To prove the decomposition of generated hydrogen peroxide to oxygen, we further compared the change of dissolved oxygen (DO) in blank experiments and NADE system. There were two blank experiments, Blank 1 was the system shown in Fig. 4a-C with almost no generation of H_2O_2 which measured the contribution of oxygen produced by the anodic oxidation to the change in DO; Blank 2 was NADE system without voltage applied which measured the contribution of naturally diffused oxygen to the change in DO. As illustrated in Fig. S11, it can be clearly seen that the concentration of DO in the NADE system continued to increase, while the blank experiments showed almost no change. These facts proved the continued decomposition of the generated H_2O_2 in the system, causing the H_2O_2 accumulation current efficiency to be less than 100%. We will improve the reactor design in the future to reduce the decomposition of H_2O_2 and improve the current efficiency of NADE.

Besides, we added these details at lines 227-233 and Fig. S10-11 in supplementary information.

1) A very high loading of catalyst (13.2 mg cm^{-2}) was used to achieve the highest current efficiency in this work, which is much higher than the typically loading. Can the author explain why such a high loading was needed to achieve a relative low H_2O_2 FE of 72.7%?

Response: We sincerely thanked the reviewer's comments. We added the H_2O_2 generation performance of NADE with low catalyst loading (1 and 2 mg cm^{-2}) and further explained the impact of the catalyst loading on the application of NADE, as written in lines 254-265.

As shown in Fig. S12, when the catalyst loading was low, NADE also had excellent H_2O_2 production efficiency and the current efficiencies were about 80% at 10 minutes. When the catalyst loading was 1 mg cm^{-2} , NADE could maintain stable H_2O_2 production efficiency (CE=63.8%-78.6%; 1 h) after running 10 times (e.g., 20 h) at the current density of 60 mA cm^{-2} (Fig. S13). However, NADE with low catalyst loading had some drawbacks when it was operated for a long time. As shown in Fig. S14a and b, after 20 h of operation, NADE with the catalyst loading of 1 mg cm^{-2} showed significant electrolyte penetration compared with NADE with the catalyst loading of 13.2 mg cm^{-2} .

For why such a high loading was needed, there are two main reasons. First, NADE relies on air to diffuse naturally through the porous carbon felt to the reaction interface. There is no artificially supplied gas on the side of the diffusion layer to maintain the pressure of the gas phase. Higher catalyst layer loading can

better prevent electrolyte leakage. The second is that higher catalyst loading can improve the stability of NADE operation at long terms of operation and high current density. For example, a satisfactory H₂O₂ production performance could be obtained at a current density of 60 mA cm⁻² when the loading of catalyst was 1 mg cm⁻², but an obvious electrolyte leakage occurred after long terms of operation. As the applied current density was increased to 200 mA cm⁻², NADE with catalyst loadings of 2 and 4.4 mg cm⁻² showed different degrees of electrolyte penetration, while NADE with catalyst loading of 8.8 mg cm⁻² did not occur at all (Fig. S14c-e). When the catalyst loading was 13.2 mg cm⁻², NADE could still maintain a high H₂O₂ production efficiency as current density reached 240 mA cm⁻², which greatly increased the application current limit of the electrode. As far as we known, there is no electrode in the literatures that can maintain high H₂O₂ production efficiency at such a high current density without artificial oxygen aeration. Therefore, it is also necessary to control the catalyst loading within an appropriate range to achieve the best performance for this novel NADE.

Compared with the literature, the catalyst loading of 13.2 mg cm⁻² in the present work is not very high. For example, Barros et al. used gas diffusion electrode (GDE) with the catalyst loading of 400 mg cm⁻² for ORR to produce H₂O₂ in reference [60]. Moreira et al. used modified GDE to increase H₂O₂ production and the catalyst loading was also 400 mg cm⁻² in reference [23]. An et al. applied the air breathing cathode for H₂O₂ electrochemical production with

the catalyst loading of 50 mg cm⁻² in supplementary reference [21]. It should also be noted that the main composite of the catalyst was carbon black (CB), which had a very low cost (¥320 / 1000 g), thus the catalyst loading of 13.2 mg cm⁻² would not increase the cost too much but keep a very stable performance.

For the relatively low H₂O₂ FE of 72.7%, it is the current efficiency at the applied current density of 60 mA cm⁻² after 2 hours of operation. It can be seen in Fig. S10 the current efficiency for H₂O₂ accumulation was 92.4% at 10 min and gradually decreased with electrolysis time. This is due to the decomposition of H₂O₂ caused by some parasitic reactions (1-3), and it will be exacerbated at high current densities.

Please see modifications in lines 254-265 in the revised manuscript, and Fig. S10, Fig. S11, Fig. S12, Fig. S13, and Fig. S14 in the Supplementary Information.

2) “As verified by LSV in Fig. 3c, a higher current response was observed upon a higher catalyst loading, means faster electron transfer in the circuit.” I guess the higher current response for higher catalyst loading sample was ascribed to the increased mass loading. I do not think this indicates the faster electron transfer. Can the author provide the mass loading normalized LSV curve?

Response: We sincerely thanked the reviewer's comments. We added error bars in Figure 3a and b and revised LSV curve of the catalyst loading of 4.4 mg cm^{-2} in Figure 3c. The mass loading normalized LSV curve was also provided below in Fig. 1, showing no big difference in current per gram of catalyst. After further analysis, we accepted the reviewer's suggestion and revised the expression. Please see modifications in line 243 in the revised manuscript.

Figure 1. Mass loading normalized LSV curve

3) At line 205, the author claimed that when the mass ratio of PTFE/CB is 0.6, the electron transfer number was calculated as 2.07, indicating an exclusive 2e-ORR process. However, the reported maximal H_2O_2 FE of the PTFE/CB-0.6 sample was around 80%. Can the author explain why?

Response: We sincerely thanked the reviewer's comments. This difference was due to the values at different conditions. The electron transfer number was

calculated as 2.07 indicating that the selectivity to generate H_2O_2 during the ORR is high and the current efficiency for H_2O_2 accumulation was nearly 100% at 10 min when the applied current density was 20 mA cm^{-2} . The electron transfer number 2.07 was calculated from a Tafel curve with an overpotential of 0.08-0.1 V. However, the current efficiency of H_2O_2 accumulation is related to the electrolysis time and the applied voltage or current. The current efficiency decreased with electrolysis time due to deleterious reaction as mentioned above. When the applied voltage and current increased, the current efficiency also decreased. As can be seen from Fig. 4d, the lower the applied current density, the higher the current efficiency after 1 hour of operation, which is closer to 90% at the current density of 20 mA cm^{-2} .

Reviewer #2:

In this paper, a new superhydrophobic carbon felt gas diffusion layer was constructed to greatly improve oxygen mass transfer efficiency. Interestingly, the natural air diffusion electrode has an oxygen utilization rate of up to 44.5%–64.9% without external aeration. In a word, the manuscript is of quality if the following comments could be addressed properly:

1. P 6, L 7, the general gas diffusion electrodes are prepared by rolling method, and the catalytic layer and diffusion layer can be tightly combined. For this NADE, does

the author evaluate the adhesion strength of catalytic layer on carbon felt substrate?
Could you please give a brief explanation about using carbon felt as the base for gas diffusion electrode? Can other porous materials be used for the substrate of NADE?

Response: We sincerely thanked the reviewer's comments. The surface of the carbon felt is rough and porous and there is good adhesion between the carbon felt substrate and the catalytic layer. In the literature, many traditional carbon felt electrodes were modified by loading similar catalysts. There are two main reasons for using carbon felt as the substrate. First, the porosity of carbon felt is more than 90%, and the resistance to gas mass transfer is very low, which is suitable for natural diffusion of air. The second is that carbon felt can become superhydrophobic and have certain strength after PTFE treatment, which can prevent electrolyte leakage.

We have also tried using other porous materials such as carbon cloth, nickel foam, aluminum foam as the substrate of NADE, but neither nickel foam nor aluminum foam was hydrophobic and the catalyst layer peeled off from these materials due to the weak adhesion to the catalyst layer.

2. P 7, L 15-17, as the author mentioned, the three-phase interfaces on catalytic layer had a short life and were easily flooded in electrolyte, an important reason is that according to the electrowetting theory, compared with the non-applied voltage, the contact angle between the solid and liquid would decrease when an external voltage was applied. How do you conclude that "after the ratio was increased to 0.6, a stable

superhydrophobic interface was formed and the three-phase interface basically didn't change with the time"? Does the author check the contact angle change of the electrode after a period of electrolysis?

Response: We sincerely thanked the reviewer's comments. We added the contact angle change of the electrode after a period of electrolysis in Fig. S6. The contact angle between the solid and liquid decreased slightly during the electrolysis process, in which the contact angle was 135.46° and 130.89° respectively after 120 minutes and 1200 minutes of electrolysis at the current density of 60 mA cm^{-2} , proving that it still maintained a very stable hydrophobic interface. We thus revised the expression in corresponding section, please see details in lines 152-154 in the revised manuscript and Fig. S6 in SI.

3. P 12, L 13-14, will the carbon felt substrate in the NADE system undergo 2-electron pathway ORR to generate H_2O_2 ? Does this contribute to the total H_2O_2 production of the electrode?

Response: We sincerely thanked the reviewer's comments. We supplemented the H_2O_2 generation performance of the carbon felt substrate without loading catalytic layer in Fig. S16. It can be seen that the H_2O_2 production was $0.92 \text{ mg cm}^{-2} \text{ h}^{-1}$ (total H_2O_2 production of the electrode was $30.94 \text{ mg h}^{-1} \text{ cm}^{-2}$ at the current density of 60 mA cm^{-2} ; P11, L5) and the current efficiency was only 2.42% (total current efficiency of the electrode was 81.3% at the current density of 60 mA cm^{-2} ; P11, L5). Therefore, the contribution of the carbon felt substrate in the

NADE system to total H₂O₂ production was very small. Please see modifications in lines 282-286 in the revised manuscript and Fig. S16 in SI.

4. P 12, L16, how do you explain the trend of H₂O₂ accumulation in Fig. 4a-C, why H₂O₂ does not seem to accumulate under this condition? Dissolved oxygen can also participate in ORR to generate H₂O₂ in situ.

Response: We sincerely thanked the reviewer's comments. The solubility of oxygen in water is very low (approximately 8 mg L⁻¹ in air atmosphere, at 1 atm and 25 °C). In a reactor without aeration, the diffusion of dissolved oxygen to the electrode surface to participate in the ORR process is very slow, so the H₂O₂ accumulation is very small. In the NADE system, the oxygen naturally diffusing to the three-phase interface from atmosphere was used to participate in the ORR process, which was the main source of oxygen reactants, but when the cathode was immersed in the electrolyte, this kind of oxygen transfer would be stopped. Therefore, the H₂O₂ accumulation in Fig. 4a-C was very small due to the low solubility of oxygen in water. We thus added this in the revised manuscript, please see details in lines 274-275.

5. P20, L 17-22, please check the layout of equations 6-9 again.

Response: We sincerely thanked the reviewer's comments. We have checked the layout of equations 6-9 to make sure they are correct.

6. In the supplementary materials, please check the number of references. Cited papers after [19] were not found in the text.

Response: We sincerely thanked the reviewer's comments. Supplementary references 20-27 were cited in section "Performance comparison with normal GDE and stability test" and Fig. 5c and d. We added the explanation at the end of supplementary information (page 30).

7. In Fig.1, the oxygen transports only through the GDE, I consider the transport pathway from the air/liquid interface maybe more important since the resistance is lower.

Response: We sincerely thanked the reviewer's comments. The ORR process mainly occurs at the gas-liquid-solid three-phase interface. NADE has excellent gas mass transfer ability, and air can diffuse through it naturally. In Fig. 4a-C, oxygen is transported through the air-liquid interface and then converted to dissolved oxygen to participate in the ORR process. The obtained H_2O_2 accumulation was very low, indicating that this contribution was very low.

Reviewer #3:

The authors present a novel natural air diffusion electrode (NADE) that shows great promise for upscaling the electrochemical production of H_2O_2 via the two-electron oxygen reduction reaction (ORR). This topic is well motivated and timely due to the

imminent need for a sustainable and decentralised alternative to the energy-intensive anthraquinone process. Experimental testing of the superior NADE performance is thorough, well-argued and overall convincing. As such, I recommend this article for publication in Nature Communications after the authors have addressed the minor revisions noted below:

1. Did the authors measure product distributions in ORR with the NADE vs the GDE? Are the <100% current efficiencies due to the competing 4e- ORR toward H₂O, or something else? I think that including this information would very much help improve this paper.

Response: We sincerely thanked the reviewer's comments. We did not measure product distributions in ORR with the NADE vs the GDE. The competing 4e- ORR toward H₂O will affect the current efficiency of hydrogen peroxide, also some deleterious reactions caused the current efficiency to decrease with time. H₂O₂ underwent its own decomposition and anode oxidation into oxygen and water. In Fig. S11, we compared the change of dissolved oxygen (DO) in blank experiments and NADE system. It can be clearly seen that the concentration of DO in the NADE system continued to increase, while the blank experiments showed almost no change. It proved that some H₂O₂ decomposition reactions in the system caused the H₂O₂ accumulation current efficiency to be less than 100%. We added the explanation in corresponding section, please see details in lines 227-233 in the revised manuscript.

2. The manuscript must be proofread carefully to correct for grammatical errors that currently make some parts of the text difficult to understand. There are quite a few such instances throughout the manuscript. A few examples are listed below:

Lines #199-201: “ This is owing to the addition of PTFE reduces the reaction site and PTFE is not electrically conductive so that excess may hinder the transmission of electrons.”

Response: We sincerely thanked the reviewer’s comments. We corrected it to “This is owing to the addition of PTFE reducing the reaction site and excess non-conductive PTFE may hinder the transmission of electrons” in lines 196-198 in the revised manuscript.

Lines #240-241: “ Electroactive Surface Area (ESA), describing the total surface which was accessible to the electrolyte and could participate in the electron transfer process.”

Response: We sincerely thanked the reviewer’s comments. We corrected it to “Electroactive Surface Area (ESA), describing the total surface that was accessible to the electrolyte and could be participated in the electron transfer process.” in lines 243-245 in the revised manuscript.

Lines #276-278: “As presented in Fig. 4c and Supplementary Fig. 10, when the diffusion area was small (0.175 cm^2), the amount of oxygen diffused to the reaction

interface by the concentration difference was very few...”

Response: We sincerely thanked the reviewer’s comments. We corrected it to “As presented in Fig. 4c and Supplementary Fig. 10, when the diffusion area was small (0.175 cm^2), oxygen naturally diffused to the reaction interface was very few...” at lines 293-295.

3. I am a bit confused about the optimum PTFE/CB ratio reported by Yu et al. (Ref. 32) as compared to the one reported in this work. If I understand correctly, Yu et al. reported an optimum PTFE/CB=5 (in fact I suggest that the authors adopt this annotation also on line #96 so that it is consistent with the presentation of their results in Figure 2). However, this work reports an optimum PTFE/CB=0.6 and suggest that higher PTFE/CB ratios result in worse performance due to particle agglomeration and sintering. Can the authors explain and comment on this difference?

Response: We sincerely thanked the reviewer’s comments. We revised the annotation in line #94 to “PTFE to CB mass ratio of 5:1”. For this difference between these two works, first of all, the electrode used by Yu et al. and the NADE used in this experiment are different in the method of loading the catalyst layer. Yu et al. immersed graphite felt in a mixture of PTFE/CB=5 and used ultrasonic impregnation to load the catalyst, but in the present study we directly coated the mixture on a carbon felt substrate. The former method required a larger proportion of PTFE in order to load the catalyst onto the graphite felt well. Second, the cathode of Yu et al. was immersed in the electrolyte, similar to the

case that described in Fig. 4a-B, and requires more PTFE ratio to maintain the hydrophobicity of the electrode during operation.

4. On line #62: I think the authors should be writing “As the applied potential increases...” rather than “As the current density increases...”

Response: We sincerely thanked the reviewer’s comments. We corrected the expression “As the current density increases...” to “As the applied potential increases...” in line 60 in the revised manuscript.

5. On line #197 the authors write “... referring to the report”. What report is this?

Response: We sincerely thanked the reviewer’s comments. We corrected the expression “referring to the report” to “according to the Tafel plots” in line 194 in the revised manuscript.

6. The abbreviation ‘EIS’ on line #208 needs to be explained.

Response: We sincerely thanked the reviewer’s comments. We added the full name explanation for the abbreviation “EIS” in line 205 in the revised manuscript.

Reviewers' Comments:

Reviewer #2:

Remarks to the Author:

This revision is satisfactory

Reviewer #3:

Remarks to the Author:

I find that the points raised during review have been satisfactorily addressed and therefore recommend publication.

Response to the reviewers' comments

Ms. Ref. No.: NCOMMS-19-38589A

Title: Ultra-efficient electrosynthesis of hydrogen peroxide on superhydrophobic three-phase interface by natural air diffusion

Reviewers' comments:

Reviewer #2:

This revision is satisfactory.

Response: We sincerely thanked the reviewer's comments.

Reviewer #3:

I find that the points raised during review have been satisfactorily addressed and therefore recommend publication.

Response: We sincerely thanked the reviewer's comments.